# The Blockade of Tumoral IL1β-Mediated Signaling in Normal Colonic Fibroblasts Sensitizes Tumor Cells to Chemotherapy and Prevents Inflammatory CAF Activation

**DOI:** 10.3390/ijms22094960

**Published:** 2021-05-07

**Authors:** Natalia Guillén Díaz-Maroto, Gemma Garcia-Vicién, Giovanna Polcaro, María Bañuls, Nerea Albert, Alberto Villanueva, David G. Molleví

**Affiliations:** 1Program Against Cancer Therapeutic Resistance (ProCURE), Catalan Institute of Oncology, 08908 L’Hospitalet de Llobregat, Spain; nguillen@iconcologia.net (N.G.D.-M.); ggarciav@idibell.cat (G.G.-V.); mbanuls@iconcologia.net (M.B.); nalbert@idibell.cat (N.A.); avillanueva@iconcologia.net (A.V.); 2Tumoral and Stromal Chemoresistance Group, Molecular Mechanisms and Experimental Therapy in Oncology Program (ONCOBELL), Institut d’Investigació Biomèdica de Bellvitge—IDIBELL, 08908 L’Hospitalet de Llobregat, Spain; 3Dipartimento Scienze e Tecnologie, Universita degli Studi del Sannio, 82100 Benevento, Italy; giovanna.polcaro@hotmail.it

**Keywords:** interleukin 1 beta, tumor microenvironment, stroma, resistance, myofibroblasts, inflammatory CAFs

## Abstract

Heterotypic interactions between newly transformed cells and normal surrounding cells define tumor’s fate in incipient carcinomas. Once homeostasis has been lost, normal resident fibroblasts become carcinoma-associated fibroblasts, conferring protumorogenic properties on these normal cells. Here we describe the IL1β-mediated interplay between cancer cells and normal colonic myofibroblasts (NCFs), which bestows differential sensitivity to cytotoxic drugs on tumor cells. We used NCFs, their conditioned media (CM), and cocultures with tumor cells to characterize the IL1β-mediated crosstalk between both cell types. We silenced IL1β in tumor cells to demonstrate that such cells do not exert an influence on NCFs inflammatory phenotype. Our results shows that IL1β is overexpressed in cocultured tumor cells. IL1β enables paracrine signaling in myofibroblasts, converting them into inflammatory-CAFs (iCAF). IL1β-stimulated-NCF-CM induces migration and differential sensitivity to oxaliplatin in colorectal tumor cells. Such chemoprotective effect has not been evidenced for TGFβ1-driven NCFs. IL1β induces the loss of a myofibroblastic phenotype in NCFs and acquisition of iCAF traits. In conclusion, IL1β-secreted by cancer cells modify surrounding normal fibroblasts to confer protumorogenic features on them, particularly tolerance to cytotoxic drugs. The use of IL1β-blocking agents might help to avoid the iCAF traits acquisition and consequently to counteract the protumorogenic actions these cells.

## 1. Introduction

During the initial steps of the tumorigenic process, newly transformed malignant cells compete with the normal microenvironment to overcome the inhibitory signals that normal tissue imposes [1]. However, once the homeostatic balance has been lost (e.g., due to cytokine disbalance) cells of the normal tissue surrounding tumors (e.g., fibroblasts, macrophages, etc.) become potent protumorigenic inducers, providing support for tumor proliferation, survival, immune evasion and invasiveness [2,3]. In a similar vein to the angiogenic switch [4], resident fibroblasts undergo a switch to become carcinoma-associated fibroblasts (CAFs). Of the various hypotheses that have been proposed [5], the most plausible is that pericryptal fibroblasts subjacent to the basement membrane enveloping epithelial glands become activated when tumor cells disrupt the basal membrane and start to invade. Such activation depends on juxtacrine and paracrine bidirectional crosstalk [6], which is mainly mediated by integrins, cadherins, soluble factors [7], and exosome cargo [8]. After epithelial malignization, pericryptal fibroblasts activate several mechanisms of innate inflammatory signals, MAP3K8 and IL1R1 being key mediators in the process [9].

A large number of soluble factors are known to be involved in this crosstalk. However, it is difficult to identify a trigger; it probably depends on multiple factors and is likely to be organ-dependent. Interleukin 1α has been proposed as leading the tumor-stroma crosstalk to sustain the expression of inflammatory mediators by CAFs in pancreatic [10] and prostate tumors [11]. Moreover, tumor-derived interleukin 1β (IL1β) has been associated with the conversion of fibroblasts into CAFs in squamous cell carcinoma [12]. In addition, blocking the interaction of IL1β with its receptor IL1R1, which is overexpressed in normal fibroblasts relative to CAFs [13,14], reprograms CAFs to hinder oral carcinogenesis [15]. Interestingly, IL1β-deficient mice do not develop tumors, unlike wild-type animals [16]. Other authors have reported that tumor cell-derived IL1α induces HGF secretion by CAFs, enhancing in turn the metastatic potential of tumor cells [17].

However, the involvement of interleukin 1 family cytokines in chemoresistance has not been fully elucidated. Mendoza-Rodriguez et al. reported that IL1β induces the overexpression of an IAP family protein (BIRC3), conferring doxorubicin resistance on breast cancer cells [18]. In addition, it has recently been reported that the blockade of stromal IL1β renders PDAC tumors less fibrotic and more sensitive to gemcitabine [19].

Several studies have examined the interaction between tumor cells and CAFs in various cancer types, but few have addressed the early steps of the heterotypic interactions between resident normal fibroblasts and tumor cells, and how the inflammatory signals provided by the tumor create a refuge to avoid the damage induced by chemotherapy. Here, we explore how cancer cell-derived IL1β induces paracrine signaling in fibroblasts that causes resistance to oxaliplatin (L-OHP; a DNA inter and intra-strand cross link inducer) in colorectal cancer cell lines.

## 2. Results

### 2.1. IL1β mRNA Expression in Colorectal Cell Lines, Fibroblasts and Cocultures

The level of IL1β mRNA expression was measured in 13 CCCL monocultures, and 11 NCFs and 15 CAFs from colorectal cancer patients. As shown in Figure 1A, levels in CCCL were highly variable, but lower than values detected in NCFs and CAFs, with mean relative expression values of 0.016 (95% CI: 0.007–0.026) in tumor cells vs. 1.05 (95% CI: 0.25–1.84) in NCFs and 2.86 (95% CI: 1.2–4.52) in CAFs. These were approximately 60 and > 150 times higher in NCFs and CAFs, respectively. However, after 5 days in transwell coculture with NCFs the mRNA expression values of DLD1 cells increased > 170-fold on average, while the increase in cocultured NCFs was more modest (2.5-fold on average), although the latter attained values were comparable to those of their paired CAFs (Figure 1B,C).

### 2.2. IL1β Receptor Expression in Colorectal Cell Lines and NCFs

Regarding the expression of the IL1β receptors, IL1R1 and IL1R2, IL1R1 was strongly expressed in fibroblasts while colorectal tumor cell lines expressed extremely low levels of the receptor (Figure 1D,E). Interestingly, the decoy receptor IL1R2 was expressed at low levels in fibroblasts, while its expression in tumor cells was variable but high in most of the CCCL tested (Figure 1F,G).

In addition, the expression of the receptors was regulated by IL1β in NHFs and NCFs, since the expression of both IL1R1 and IL1R2 increased after stimulation with IL1β, while receptors in tumor cells (SW480 and DLD1) were not affected by IL1β, probably due to the high basal levels of expression of the decoy receptor IL1R2 in those cancer cells (Figure 1H).

### 2.3. Differential Impact of IL1β on the Proliferation of Tumor Cells and Fibroblasts

The aforementioned results imply that IL1β might act first in a paracrine manner from tumor cells to myofibroblasts, and that an autocrine loop might maintain the activated fibroblast phenotype after fibroblast activation. In fact, we observed that, after IL1β stimulation, NCFs lost their myofibroblastic features (αSMA, Calponin, and Synaptopodin expression) and overexpressed FAP (Figure 2A). We first checked whether proliferation might be stimulated after culturing cells, CCCL and fibroblasts, with (10 ng/mL) of IL1β. As shown in Figure 2B, 5 out of 7 colorectal cells lines showed no increase in proliferation after exposure to IL1β. Conversely, NCFs and NHFs both showed increased proliferation after (10 ng/mL) IL1β stimulation. This proliferation is controlled by blocking the action of IL1β, either by acting on one of the most important effectors of the pathway, P38 by the addition of a P38 inhibitor VX-702 (400 nM), or by altering the binding of the cytokine with its receptor with a neutralizing IL1β antibody (Figure 2C).

Using the two cell lines that responded to IL1β, we conducted a dose–response assay, in which HCT116 increased proliferation according to the increase in IL1β (Figure 2D; statistically significant only at 10 ng/mL IL1β), while HT29 slightly increased proliferation but was not associated with an increase in IL1β (Figure 2E). However, this increase was not associated with an increase in viability after culturing cells with 2 µM L-OHP.

### 2.4. Exploring the Effect of IL1β and Its Targets on Fibroblast and Tumor Cell Migration and Recruitment

We explored other protumorogenic properties, such as migration, on which IL1β and their soluble targets could be altering the crosstalk between tumor cells and fibroblasts. Fibroblast migration was stimulated after IL1β administration, both, in a wound-healing assay (Figure 2F; *p* = 0.0035 Kruskal–Wallis test, with Dunn’s *post hoc* multiple comparison test) and directional migration using transwell inserts (Figure 2H; adjusted *p* = 0.0034 Kruskal–Wallis test, with Dunn’s *post hoc* multiple comparison test), while DLD1 cells did not increase migration as expected. When we used CM from IL1β-stimulated NCF (48 h of fibroblast stimulation), IL1β targets secreted into the culture medium from NCFs enhanced the migratory capabilities of tumor cells to an almost statistically significant extent relative to control or NCF CM (Kruskal–Wallis test, with Dunn’s *post hoc* multiple comparison test, adjusted *p* = 0.051; Figure 2G). Directional migration of DLD1 cells in coculture with NCFs was also diminished with the use of neutralizing antibody against IL1β (*p* = 0.0006, U Mann–Whitney test; Figure 2I). Thus, blocking the action of IL1β disrupted the migratory capabilities of tumor cells and restored the myofibroblastic phenotype of NCFs as observed at the protein level for FAP and αSMA (Figure 2J).

### 2.5. Chemoresistance Induced by IL1β-Responsive Soluble Factors

We checked the effect of IL1β using dose–response curve assays against increasing doses of L-OHP. Once again, only HT29 cells responded to IL1β, inducing a significant shift in the IC_50_ curves for L-OHP (Figure 3A). None of the other CCCL were chemoprotected from L-OHP by IL1β; however, we explored whether the secretion of IL1β-responsive proteins into the medium, could be responsible for inducing a protective milieu for tumor cells, thereby increasing the viability of tumor cells. As illustrated in Figure 3B, IL1β-stimulated NCF-CM stimulated proliferation of all the tumor cells tested (dark grey bars, left vertical axis in Figure 3B) but also promoted the viability of CRC cell lines exposed at fixed doses of L-OHP (dark grey bars, right vertical axis, Figure 3B; significant P values correspond to adjusted P values after Dunn’s multiple comparison test). Interestingly, both proliferation and viability could be restored to the same levels as non-stimulated NCF-CM (black bars in Figure 3B) by the addition of an IL1β-blocking antibody to the IL1β-stimulated NCF cultures (light grey bars, Figure 3B). The observation was reproduced when assaying the protective effect of IL1β targets in a dose–response assay (Figure 3C, left panel), in which the blocking antibody neutralized the effect of IL1β, thereby avoiding the secretion of IL1β targets into the CM, and restoring the IC_50_ values for L-OHP in DLD1 cells. In addition, the protective effect induced by fibroblast-secreted products was not restricted to stimulated NCFs since normal foreskin fibroblasts stimulated with IL1β yielded the same observation (Figure 3C, middle and right graphs). We can not attribute such protective effect to a single IL1β target, since as described in Appendix A, many of the assessed soluble factors are known chemoprotective cytokines and chemokines as IL6, IL8, CCL2 among others (Appendix A).

To demonstrate that the crosstalk between tumor cells and fibroblasts mediated by IL1β induces chemoresistance, we performed a colony forming assay in transwell cocultures between different CCCL (700 cells per well; lower chamber) and NCF (50,000 cells, 24 mm inserts 0.4 nm pore size), with the addition of L-OHP. To hamper the action of IL1β, a blocking IL1β antibody had been added to the test group. After 10 days in coculture we stained the lower chambers with crystal violet. As depicted in Figure 4A,B, cocultures with the addition of the neutralizing antibody displayed a smaller number of colonies for all the cell lines tested. To evaluate the effectiveness of the addition of the blocking antibody, we assessed the presence of free IL1β in the coculture pooled supernatant (equal volumes from cocultures with DLD1, HCT116, or HT29), as well as IL6, as a surrogate marker of the IL1β response, observing that the addition of the neutralizing antibody blocked both IL1β and IL6 (Figure 4C).

As expected, most of the IL1β targets stimulated JAK/STAT and AKT pathways in tumors cells. This activation can be reversed by neutralizing the binding of IL1β to their receptors in NCFs (Figure 4D). In addition, CM from IL1β-treated fibroblasts induces the overexpression of Cyclin D1 and cMyc, which might help explain how IL1β-soluble targets influence the cell-cycle progression and chemoresistance observed in tumor cells, when treated with L-OHP (Figure 4E).

To find out if the source of IL1β that stimulate the fibroblasts in a paracrine form could be the interleukin generated by the tumor cells, we silenced the expression of IL1β in the HT29 cells (Appendix A). Thus, if the tumor cells, in response to soluble factors of the NCFs, could not secrete IL1β, the fibroblasts would not respond with the secretion of cytokines that altered the damage induced by the chemotherapy (experiment overview in Figure 4F). Therefore, the conditioned media generated from IL1β-silenced HT29 cells and fibroblasts cocultures, when used to stimulate HT29 cells in a dose response assay against L-OHP, protected to a lesser extent against the drug (Figure 4G, light grey line), compared to conditioned media from cocultures obtained with wild-type and non-silenced (mock) HT29 cells and fibroblasts. The exogenous addition of recombinant IL1β to the IL1β-silenced HT29 cocultures restored the IL1β targets, and consequently the IC_50_ values of HT29 cells to L-OHP (Figure 4E). Likewise, when we explored the effects of IL1β-silenced tumor cells on co-cultured NCFs, we could observe that cocultured fibroblasts lost the expression of inflammatory cytokines, as IL6, LIF or CCL2, while acquired markers of myofibroblasts (ACTA2, PDPN, MYH11, or CNN1; Figure 4H).

This fact, together with observations in Figure 2A,J let us to hypothesize that perhaps the population of inflammatory CAFs could be responsible of altering the chemosensitivity against cytotoxic drugs like L-OHP, while myofibroblastic CAFs might have other functions. To test this, we stimulated NCFs with (10 ng/mL) IL1β or (20 ng/mL) TGFβ1 for 48 h. With the conditioned media we performed a dose–response assay against L-OHP in CCCL. As illustrated in Figure 5A, only the IL1β-stimulated NCFs CM increased the L-OHP IC_50_ values (Fisher test; *p* < 0.0001), while the TGFβ1-stimulated NCFs CM did not differed from the NCF CM control, both in DLD1, HT29, and HCT116 cells. Similar results were observed in a colony forming assay, where IL1β-treated NCF conditioned medium significantly avoided the reduction of colonies induced by 5 µM and 10 µM L-OHP than TGFβ1-treated NCF conditioned medium (adj *p* = 0.013 and adj *p* = 0.005, respectively; Kruskal–Wallis test with Dunn’s correction for multiple comparison; Figure 5B).

Real-Time PCR assessment of CAF’s markers on treated NCF shows that, in response to IL1β there is a statistically significant increase of FAP, IL1β, IL6, and CCL2, markers associated with an inflammatory CAF phenotype (iCAF), being all the cytokines tested involved with the induction of chemoresistance-associated pathways. TGFβ1-treated NCFs displayed, as expected, a more myofibroblastic phenotype. However, CLEC3B and GSN, genes reported to be markers for iCAF, were expressed as TGFβ1 targets (CLEC3B) or no difference between both stimulations (GSN).

## 3. Discussion

The interplay between tumor cells and their neighboring normal cells occurs even at early stages of tumorigenesis, when few transformed epithelial cells interact with pericryptal myofibroblasts. These fibroblasts play an important role in regulating the normal colorectal stem cell niche, controlling normal tissue homeostasis and facilitating tumor progression when homeostasis is lost [21], and is probably the main source of CAFs after appropriate stimulation. 

A wide range of cytokines, growth factors have been reported as having a role in the differentiation of fibroblasts into myofibroblasts, like TGFβ1 [22], PDGFβ [23], IL13 [24], etc. Conversely, IL1β has been described as inhibiting myofibroblast differentiation, decreasing the levels of αSMA and other contractile proteins [25]. Nevertheless, the role of IL1β in the context of the differentiation of myofibroblasts to CAFs has been less thoroughly studied, particularly with respect to the CAF subtypes in which differential αSMA expression has recently been reported [26,27].

IL-1 expression is elevated in several human cancers, and patients with tumors that express IL-1 generally have a worse prognosis for their disease. In fact, the involvement of IL-1 and particularly IL1β in proliferation, invasion and metastasis has long been known. Two studies in animals in the early 1990s showed that a single dose of IL-1 administered just before the intravenous injection of tumor cells increased the number of lung metastases [28,29]. Paracrine crosstalk mediated by tumor-derived IL1β and CAFs was also described [12]. Moreover, such crosstalk seems to be very important in incipient neoplasia to orchestrate tumor-promoting inflammation [30]. However, the involvement of IL1β, particularly tumor cell IL1β crosstalk with normal resident myofibroblasts, either at a primary site (NCFs) or at a distant niche (NHFs), altering the sensitivity to cytotoxic chemotherapy has not been described. Here, we report a IL1β-mediated paracrine loop from tumor cells towards normal resident fibroblasts, that elicits a secretory response in fibroblasts that increases the tolerance of tumor cells to L-OHP and might be responsible for the pool of the iCAFs subset. Such a process might be altered by the addition of an IL1β-blocking agent, rendering tumor cells without the protection conferred by fibroblast-soluble factors, and avoiding NCF-to-CAF conversion. 

The autocrine role of IL1β in inducing chemoresistance in pancreatic cancer cells was first demonstrated in 2002 [31]. In our experiments, we did not observe any direct action of IL1β on tumor cells, similar to results published by Young H et al. [32] for BRAF and MEK inhibitors. We concluded that IL1β did not by itself induce resistance to conventional drugs used in CRC treatment. In fact, we have shown how IL1β secreted by cancer cells is used to educate surrounding normal resident fibroblasts to create a niche that first induces the recruitment of fibroblasts (enhanced migration) and their proliferation, and then helps tumor cells to overcome the damage induced by chemotherapeutic cytotoxic agents, such as L-OHP and 5-fluorouracyl. In this context, this crosstalk contributes to induce a certain tolerance to chemotherapy-induced damage, a transient protection, until the tumor cells are capable of developing resistant phenotypes. This fact is not limited to chemotherapeutic drugs since a similar crosstalk induced by macrophage-derived IL1β that generates tolerance to BRAF and MEK inhibitors has been published [32]. In particular, these authors observed that IL1β secreted by cancer-associated macrophages induced the secretion of CXCR2 ligands by CAFs, such as IL8 and GROα (CXCL1), which were ultimately the mediators of such tolerance in tumor cells. In fact, IL8, CXCL1, along with IL6 and CCL2, are the most represented soluble molecules in the conditioned media of IL1β-treated NCF, out of the 174 tested molecules (Appendix A). 

It is also of particular note that, in the context of the recently described coexistence of different CAF subpopulations [27,33,34], IL1β seems to downregulate the myofibroblastic phenotype to potentiate the FAP-positive inflammatory phenotype (iCAF), a process that could be avoided by blocking the action of IL1β. Thus, in an indirect process, IL1β gives rise to the differential sensitivity of CRC cancer cells.

To examine the signaling events that occur in tumor cells after stimulation by CM, we set our sights on signaling pathways in which we had previously observed involvement in de novo resistant processes [35]. As expected, IL1β-soluble targets mentioned above, like IL8, IL6, and CCL2, activated the JAK/STAT and PI3KCA/AKT pathways. Hindering the interplay between IL1β and its receptors in myofibroblasts renders these cells less active and less proficient at secreting protumoral and chemoprotective cytokines and growth factors. In turn, tumoral cells showed decreased levels of activation in both pathways. This raises the possibility of using anti-IL1β agents as a coadjuvant treatment in combination with cytotoxic drugs to avoid environmentally mediated drug-resistant processes. Furthermore, blocking IL1β might help in the clinical control of therapy-induced inflammation that could impede the effectiveness of chemotherapy [36]. In fact, the utility of canakinumab (anti-IL1β monoclonal antibody) has been explored in two clinical trials in solid tumors (NCT03447769, NCT02900664). Other agents used to treat rheumatoid arthritis, like anakinra (an IL1-receptor antagonist), have been repositioned as an anti-cancer treatment in PDAC, metastatic CRC (NCT02550327, NCT02090101). 

Our results, in the context of colorectal cancer management, suggest a way of increasing the response rate to the neoadjuvant treatment in rectal cancer, the only CRC setting in which primary tumors are treated before surgery. This might help to restrain the local tumor’s growth, avoiding paracrine signals from the localized tumor to adjacent tissue and then blocking tumor-induced and therapy-induced proinflammatory signals that favor chemoresistance. In fact, it has recently been demonstrated that normal adjacent colonic tissue can predict prognosis in colorectal cancer [37]. Such an aberrant transcriptomic profile of supposedly normal tissue is the consequence of the crosstalk with the tumor [38]. However, it is unclear what is activating what. In other words, which factor causes the tumor cells’ increased expression of IL1β? This question, for the moment, is difficult to answer. We explored CCL11 and CCL2, the two most highly expressed cytokines of the 174 soluble factors tested, by NCFs under basal conditions. Neither of them caused overexpression of IL1β in CRC cell lines. Other less well represented soluble factors, or a combination of different cytokines and growth factors, might be the trigger.

In recent years there has been an explosion in terms of the appearance of countless scientific articles describing different CAFs subsets [20,39,40], although most of them could be summarized in two main classes, pro-inflammatory iCAFs and others with a purely myofibroblastic profile, myCAFs, characterized by the production of collagens and other proteins of the extracellular matrix. The protumoral or antitumor role of each of these subsets described is not clear. Both, IL1β and TGFβ1, along with some more soluble factors, have been described to be molecules that mediate the crosstalk between tumor cells and fibroblasts and are responsible for the activation state of the latter [22,41]. However, different ligands will induce different CAF subsets. Recently it has been described that IL1β and TGFβ1 appear to act antagonistically [42]. This fact contributes to the plasticity of fibroblasts depending on their topographic location as the tumor modifies its shape with growth. However, our results contribute to elucidate which subset is more suitable to be eliminated or modified therapeutically, since iCAFs can contribute to the development of chemotherapy-resistant phenotypes. In any case, we must be cautious in this regard since it has been shown that the use of JAK inhibitors to attenuate the pro-inflammatory response has increased the proportion of myofibroblasts, with the consequent increase in extracellular matrix deposits, which end up hindering the access of chemotherapy into the tumor [43,44]. This fact would support the strategy of combining an anti-iCAF treatment together with modulation of the action of TGFβ1, in a similar way to what our group published a few years ago by combining a TAK1 inhibitor in combination with Galunisertib [41].

In summary, the soluble factors produced by normal colonic fibroblasts in response to IL1β generate chemoresistance in CRC cells, while they do not produce such an effect in response to TGFβ1. Hindering IL1β-mediated crosstalk between fibroblasts and tumor cells might counteract inflammatory signaling, causing de novo resistance. This raises the possibility of using anti-IL1β to prevent such tumor-mediated and therapy-induced proinflammatory events that sensitize tumors to chemotherapy. Such agents, in combination with anti-TGFβ1 therapies, might block the recruitment of new fibroblasts into the growing tumor, helping to restrain the cancer and avoid new stroma activation.

## 4. Methods

### 4.1. Culture of Primary Fibroblasts and Preparation of Conditioned Medium (CM)

Fresh surgical specimens were obtained with the approval of the Ethics Committee of the Hospital Universitari de Bellvitge (IDIBELL), under patient’s informed consent (approval reference PR91/10, IDIBELL Ethics Committee, 6 May 2010). Tissue samples from morphologically normal colonic mucosa (at least 5 cm from the tumor’s surgical margin), primary tumors and normal liver (at least 5 cm from the resected liver metastasis) were minced and incubated with collagenase (Stemcell technologies, Saint Égrève, France) and dispase (1 U/mL in DMEM/F12; Stemcell technologies, Saint Égrève, France) for 2 h at 37 °C. Cells were resuspended and plated with Dulbecco’s modified Eagle’s medium-F12 (DMEM/F12, Gibco, Life Technologies, Oslo, Norway) containing 10% fetal bovine serum (FBS; Gibco, Life Technologies, Oslo, Norway) and penicillin/streptomycin antibiotics. Primary cultures from normal colonic fibroblasts (NCFs), paired CAFs, or normal hepatic fibroblasts (NHFs) were established and routinely maintained at 37 °C in a humidified atmosphere containing 5% CO_2_. After a maximum of 4 passages, RNA and protein were obtained to check for fibroblast purity. To obtain conditioned media (CM), 10^6^ NCFs were incubated for 48 h in 10 cm diameter dishes in DMEM/F12 or DMEM/F12 + IL1β (10 ng/mL). CM was collected, centrifuged for 5 min at 3000 rpm to remove cell debris, sterile-filtered through 0.22-μm filter units (Millex^®^ GS, Millipore), and stored at −80 °C until use.

### 4.2. Colorectal Cancer Cell Lines

DLD1, LoVo, Colo-205, RKO, Co115, HCT-15, KM12C, Caco-2, HCT116, HT-29, SW480, SW620, and SW1116 were all cultured in 10% FBS DMEMF12. Cells were periodically tested for mycoplasma contamination and were authenticated by short tandem repeat profiling. For functional assays, we have used the most appropriate cell lines in each particular experiment for the correct visualization and execution of the designed objectives in concordance with our previous experience with the aforementioned cell lines.

### 4.3. Cocultures of NCFs with Colorectal Cancer Cell Lines (CCCL)

Isolated NCFs were cocultured with DLD1 cells for 5 days in 6-well transwell inserts 3 µm pore-size. After coculturing, total RNA was extracted for IL1β mRNA quantification by qRT-PCR. Unless stated, fibroblasts were plated in the lower chamber (10^5^ cells) and tumor cells in the upper chamber of the insert (10^4^ cells).

In specific co-culture experiments, HCT116 and HT29 were also used.

### 4.4. RNA Isolation and Quantitative Real-Time PCR (qRT-PCR)

Total RNA from different isolated fibroblasts and CCCL was extracted using the TRIzol^®^ reagent method and column purification using PureLink^TM^ RNA Mini Kit (Invitrogen, Van Allen Way, Carlsbad, California 92008, USA). RNA quantity was determined with a NanoDrop ND-1000 spectrophotometer (NanoDrop Technologies Inc., Rockland, DE, USA) and 100ng of total RNA was reverse-transcribed using M-MLV reverse transcriptase (Invitrogen), following the manufacturer’s instructions. A 0.1-µg equivalent of the corresponding cDNA was used for each quantitative PCR assay performed with the LightCycler^®^ 480, SYBR Green I Master (Roche Applied Science, Mannheim, Germany). Primers for IL1β, IL1R1, IL1R2, ACTA2, FAP, IL6, COL3A1, COL1A1, MYH11, PDPN, S100A4, CCL2, CNN1, LIF, CLEC3B, and GSN were designed using Primer3 Input (https://primer3.ut.ee/, accessed on 12 June 2019), and predicted PCR product sequences were verified using BLAST (http://www.ncbi.nlm.nih.gov/blast, accessed on 12 June 2019). For all genes tested, annealing temperature was 60 °C and 40 cycles were performed. All primer sequences are available upon request. In all cases, GAPDH was used for normalizing expression values.

### 4.5. Cell Proliferation and Viability

The WST-1 proliferation assay was performed following the manufacturer’s instructions (Roche). Previously, 2000 cells/well (CCCL) or 20,000 cells/well (fibroblasts) were plated in six replicative wells in 96-well plates and allowed to attach, then grown in DMEM/F12 at 37 °C overnight. Next day, treatments were added accordingly, depending on which specific experiments, detailed in the corresponding figure legends, were to be carried out. The cells were incubated for 5 days. When cells were treated in dose–response curves against (L-OHP), the effect of the drug on each cell line in the presence or absence of CM or IL1β was calculated by normalizing the number of cells after 5 days of continuous treatment to the maximum number of cells in each treatment. 

### 4.6. Wound-Healing Assay and Directional Migration

Migration of cancer cells and CAFs was measured by a wound-healing and transwell assays. For wound-healing, cells were seeded in 6-cm diameter plates and cultured until confluent. Cells were starved for 24 h, and then the cell monolayer was scratched with a yellow 200-µL pipette tip to create a wound. After several PBS (phosphate-buffered saline 1×) washes to remove floating cells, for the DLD1 cell migration assay, FBS-free conditioned medium from NCF or IL1β-treated NCF or DMEM/F12 (control) was added. For the NCF migration assay, IL1β (10 ng/mL) or IL1β plus IL1β neutralizing antibody (R&D Systems AF-201-NA) were added (2 µg/mL). Pictures were taken at different times. The area between cell front margins was measured with Leica software (Wetzal, Germany) in three replicates. For directional migration, transwell PET inserts (12 mm, 8 µM pore size) were used. For NCF migration, 10^4^ cells were plated in the upper chamber. After twelve hours, FBS was removed and IL1β or anti-IL1β neutralizing antibody was added accordingly. For tumor cell migration in coculture, confluent NCFs were grown in the lower chamber and 5 × 10^3^ DLD1 cells were plated in the upper chamber. Anti-IL1β neutralizing antibody was added in the test group wells. After 24 h, cells were stained with crystal violet and transwell membranes were mounted on slides. Four microphotographies were taken for well and cells were counted manually with imaging software.

### 4.7. Silencing IL1β by Lentiviral shRNA

GIPZ lentiviral shRNA (cat #V3LHS_321411 and #V3LHS_321412 clones purchased from Dharmacon; Lafayette, CO, USA) was used to establish IL1β knockdown in HT29 colorectal cell lines. Glycerol-stored plasmid was previously replicated, isolated, transfected in HEK293T cells, and finally, transduced in target cells. Cells were selected with puromycin (2 μg/mL), according to kill curve values. A non-silencing GIPZ lentiviral shRNA containing a random vector was used as a control in subsequent experiments. Real-time PCR was performed to confirm IL1β silencing efficiency in HT29 cells.

### 4.8. Cytokines and Chemicals

Human recombinant IL1β and TGFβ1 were purchased from Peprotech (ref 200-01B, lot 1202B95R1 and 100-21C lot 0312354-1, respectively) and always used at (10 ng/mL) and (20 ng/mL), respectively. Neutralizing antibody against IL1β was purchased from R&D Systems (reference AB201-NA, lot AM1311041). VX-702 (P38 inhibitor) was purchased from Selleck Chemicals (Houston, TX, USA) and dissolved in DMSO as a stock solution and used at (400 nM). L-OHP was obtained from our own hospital Pharmacy Department in 5 mg/mL vials and diluted at working dilutions with sterile water. 

### 4.9. Western Blot Analysis

Protein concentration was determined by the BCA Protein Assay (Pierce, Rockford, IL, USA). 30 µg of the protein extract was subjected to sodium dodecyl sulfate polyacrylamide gel electrophoresis (SDS-PAGE) and transferred to PVDF membranes. After blocking for 1 h with 5% dried non-fat milk in TBS 1 × Tween 0.1% (or 5% BSA for phosphoproteins), the membranes were incubated with primary antibody (all obtained from Cell Signaling Technology, Danvers, MA, USA, except αSMA, Dako M0851, FAP, SCBT sc-100528, Calponin SCBT sc-58707, Synpo2 Sigma HPA030665) diluted 1:1000 in 1% bovine serum albumin (BSA) in TBS 1× Tween 0.1%. Antibody binding was detected using a secondary antibody diluted 1:2000 in TBS 1× Tween 0.1% and an enhanced chemiluminescence (ECL) detection kit (Amersham plc, Amersham, Buckinghamshire, UK). α-tubulin expression was used as an endogenous control.

### 4.10. Cytokine Arrays and ELISA

CM (1% FBS) from NCF or ILβ-treated NCF or IL1β-treated NCF plus neutralizing IL1β antibody was used for hybridization on the glass-slide human cytokine array G2000 from RayBiotech (Norcross, GA, USA), which detects 174 human cytokines. Samples were processed and analyzed by Tebu-Bio (Le Perray-en-Yvelines, France).

IL6 and IL1β were determined in supernatants obtained from cocultures CM by means of ELISA (R&D cat number D6050 and DLB50, respectively).

### 4.11. Statistical Analysis

Data from four independent experiments were analyzed using the nonparametric two-sided Mann–Whitney U test to compare group differences between experimental and control samples, with significance being concluded for values of *p* < 0.05. In experiments with three or more groups, comparisons were made with the Kruskal–Wallis test and Dunn’s multiple comparison test reporting adjusted P value after multiple testing. Bar graphs depicted mean values plus standard deviation. In dose–response curves, an extra sum-of-squares F test was carried out in GraphPad PRISM 8 (GraphPad Software, San Diego, California, USA) to identify statistically significant differences (*p* < 0.05) between IC_50_ curves. Sample size calculation was estimated using the web tool https://www.imim.es/ofertadeserveis/software-public/granmo/ (accessed on 7 May 2021), using type I alpha error of 0.05, two-tailed test, and beta error 0.2.

## Figures and Tables

**Figure 1 ijms-22-04960-f001:**
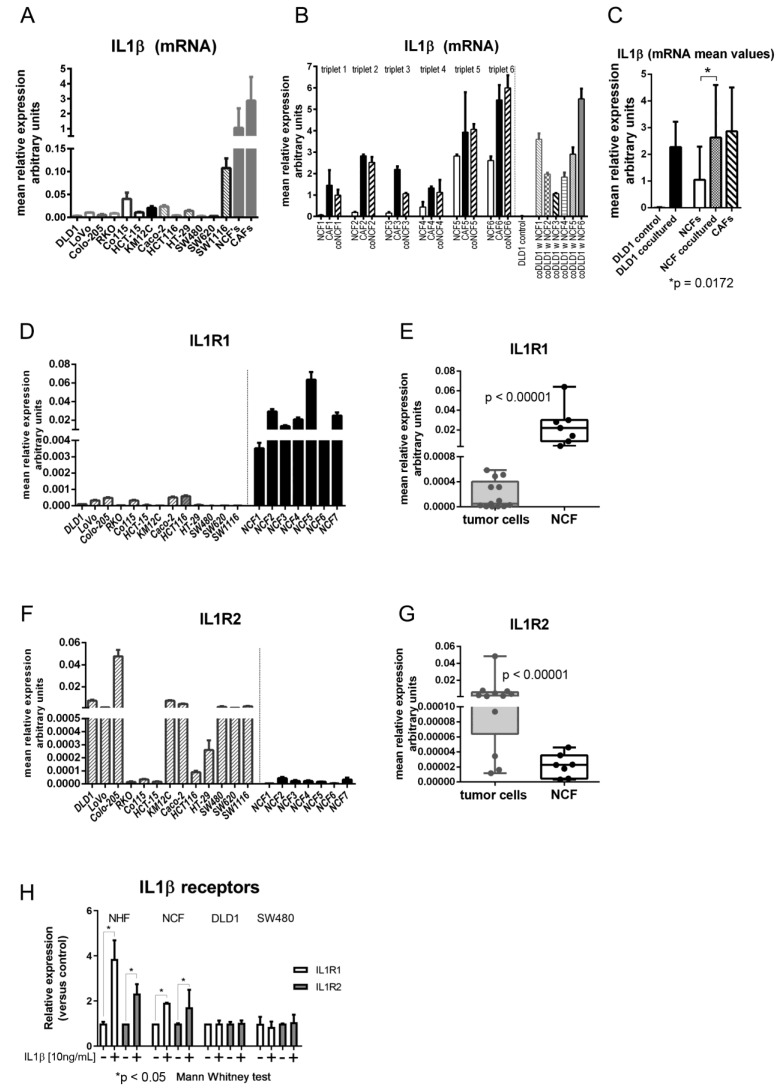
(**A**) mRNA levels of IL1β in 13 colorectal cell lines (DLD1, LoVo, Colo-205, RKO, Co115, HCT-15, KM12C, Caco-2, HCT116, HT-29, SW480, SW620, and SW1116) and in a mean of 11 normal colonic fibroblasts (NCFs) and a mean of 15 carcinoma-associated fibroblasts (CAFs). Bars depicted mean + sd of four independent biological replicates of three technical replicates each. Expression is reported as relative values corrected by housekeeping gene expression (GAPDH). (**B**) mRNA levels of IL1β in NCFs (cultured alone; depicted as white bars), paired CAFs (from the same patient; black bars) and the NCFs cultured with DLD1 cells (in transwell inserts; dashed bars). Thus, we had six different triplets, consisting in NCF and CAF from the same patient, and the NCF cocultured with DLD1 cells. In addition, we show IL1β mRNA levels in cocultured DLD1 cells with each of the 6 NCFs, compared with the DLD1-monocultured controls. Bars depicted mean + sd of four independent biological replicates. Expression is reported as relative values corrected by housekeeping gene expression (GAPDH). (**C**) mean values of mRNA IL1β. After coculturing, DLD1 cells attain values like those in NCFs and CAFs. Cocultured NCFs also increase mRNA levels to the same values as CAFs (Mann–Whitney U test; Expression is reported as relative values corrected by housekeeping gene expression (GAPDH). (**D**) mRNA relative levels of IL1R1 receptor (in relation to housekeeping gene GAPDH) in 13 colorectal cell lines and in 7 NCFs. (**E**) Mean values of IL1R1 between colorectal cell lines and NCFs are significantly different (*p* < 0.00001; Mann–Whitney U test). (**F**) mRNA levels of IL1R2 decoy receptor in 13 colorectal cell lines and in 7 NCFs. (**G**) Mean values of IL1R2 between colorectal cell lines and NCFs are significantly different (*p* < 0.00001; Mann–Whitney U test). (**H**) mRNA levels of IL1R1 and IL1R2 increase after stimulation with IL1β in normal hepatic fibroblasts (NHFs) and NCFs. Conversely, stimulation in tumor cells produced no increase in either IL1β receptor. Bars depicted mean + sd of three independent replicates. Expression values adjusted by housekeeping gene expression (GAPDH). Data were normalized to each respective control without IL1β.

**Figure 2 ijms-22-04960-f002:**
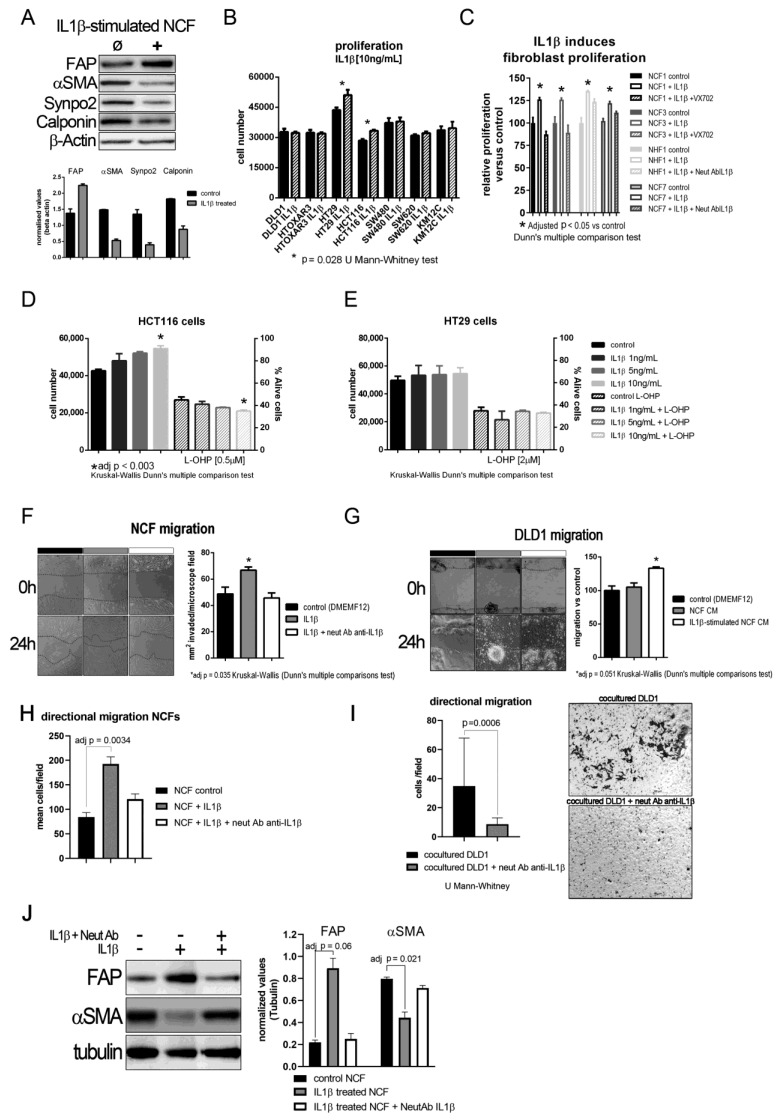
(**A**) Representative western blot of phenotypic changes observed in NCF stimulated with IL1β. After 72 h IL1β-stimulation (10 ng/mL) NCF myofibroblasts lose myofibroblastic markers (αSMA, Calponin, Synpo2) and overexpress activated CAF marker fibroblast activating protein (FAP). The bar graph below depicted mean (plus standard deviation) normalized densities (using β-Actin as loading charge) for three independent western blot cell extracts, illustrating the decrease in myofibroblastic markers αSMA, Calponin, and Synaptopodin 2. (**B**) IL1β (10 ng/mL) induced tumor cell proliferation in only two of the seven colorectal cancer cell lines. Bars depicted mean + sd of four independent experiments of three replicates each. (**C**) Conversely, IL1β induced proliferation of NCFs in a 5-day WST-1 assay. These values could be restored by the addition of a P38 inhibitor (VX-702, 400 nM) and a neutralizing polyclonal antibody against IL1β (2 µg/mL) (Kruskal–Wallis, Dunn’s multiple comparison test; bars depicted mean + sd of four independent experiments of three technical replicates each). Selecting the cell lines that responded to (10 ng/mL) IL1β, we checked the dose–response effect of IL1β on proliferation and survival against IC_50_ values for oxaliplatin (L-OHP), observing a dose–response trend only in HCT116 cells (**D**). IL1β did not induce any protection against L-OHP. No effect was observed in HT29 cells (**E**). Both D and E represent mean values + sd of three independent experiments of six technical replicates each. Assaying the effect of IL1β on fibroblast migration revealed a statistically significant increase in migration induced by the interleukin (**F**). We reported the same observation when analyzing directional migration of fibroblasts (**H**). This effect could be counterbalanced by the addition of a neutralizing polyclonal antibody against IL1β (2 µg/mL). Bars displayed mean values + sd of three independent experiments. Conditioned media from IL1β-stimulated NCFs induced the migration of tumor cells, both in a wound healing assay (**G**); white bar; bars displayed mean values + sd of three independent experiments; adj *p* value = 0.051) and directional migration in transwell, seeding NCFs in the bottom chamber (**I**); *p* = 0.0006, U Mann–Whitney test). (**J**) Representative western blot displaying that the blocking of IL1β with a polyclonal neutralizing antibody anti-IL1β (2 µg/mL) maintains the myofibroblastic phenotype in NCFs determined as the expression of αSMA and the decrease of FAP. The bar graph shows the mean + sd for three independent experiments not reaching statistical significance for FAP (*p* = 0.06 after adjusting for multiple comparison), but significant for αSMA (*p* = 0.021, after adjusting for multiple comparison; Kruskal–Wallis plus Dunn’s multiple comparison test).

**Figure 3 ijms-22-04960-f003:**
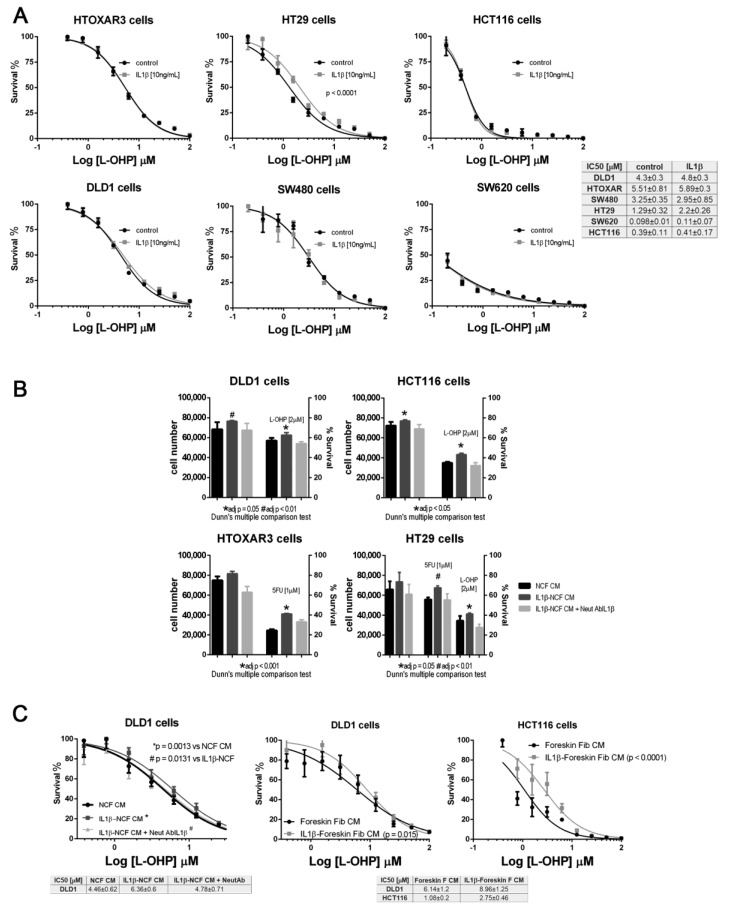
(**A**) Panel with dose–response curves for L-OHP of six colorectal cancer cell lines cultured under standard conditions (black lines) or in the presence of (10 ng/mL) IL1β (grey lines). IL1β displaced the IC_50_ values for L-OHP only in HT29 cells. Each dose–response curve corresponds to the mean of three independent experiments of six technical replicates each. Differences between mean (*n* = 3) dose–response curves were compared with extra sum-of-squares F test (Log IC_50_). Survival is reported as %. (**B**) Panel of four colorectal cell lines treated with conditioned medium (CM) from NCFs (black bars), IL1β-stimulated NCFs (dark grey bars), or IL1β-stimulated NCFs plus a neutralizing antibody against IL1β ((2 µg/mL); light grey bars). IL1β used at (10 ng/mL). For all cell lines tested, IL1β-stimulated NCFs CM promoted proliferation (left Y axis) of tumor cells, although the effect relative to control NCFs CM was only statistically significant in DLD1 and HCT116 cells. The right Y axis shows that, for all cell lines and drugs (L-OHP and 5FU), the viability of cells cultured with IL1β-stimulated NCFs CM was greater than that of controls (Kruskal–Wallis, Dunn’s multiple comparison test, adjusted P values), meaning that IL1β targets modified the sensitivity to both drugs. Such sensitivity was restored by the addition of a neutralizing IL1β antibody during NCF culture for CM production. Bars depicted mean + sd of four independent experiments of six technical replicates each. (**C**) The same observation as described in (**B**), in dose–response curves, where IL1β-stimulated (10 ng/mL). NCFs CM induced a shift in the IC_50_ curves for L-OHP, leading to an increase in tolerance of cytotoxic compounds (left plot). Similar results were obtained for DLD1 and HCT116 using foreskin fibroblasts (middle and right graphs). Survival is reported as %. In both cases, dose–response curves correspond to the mean of three independent experiments.

**Figure 4 ijms-22-04960-f004:**
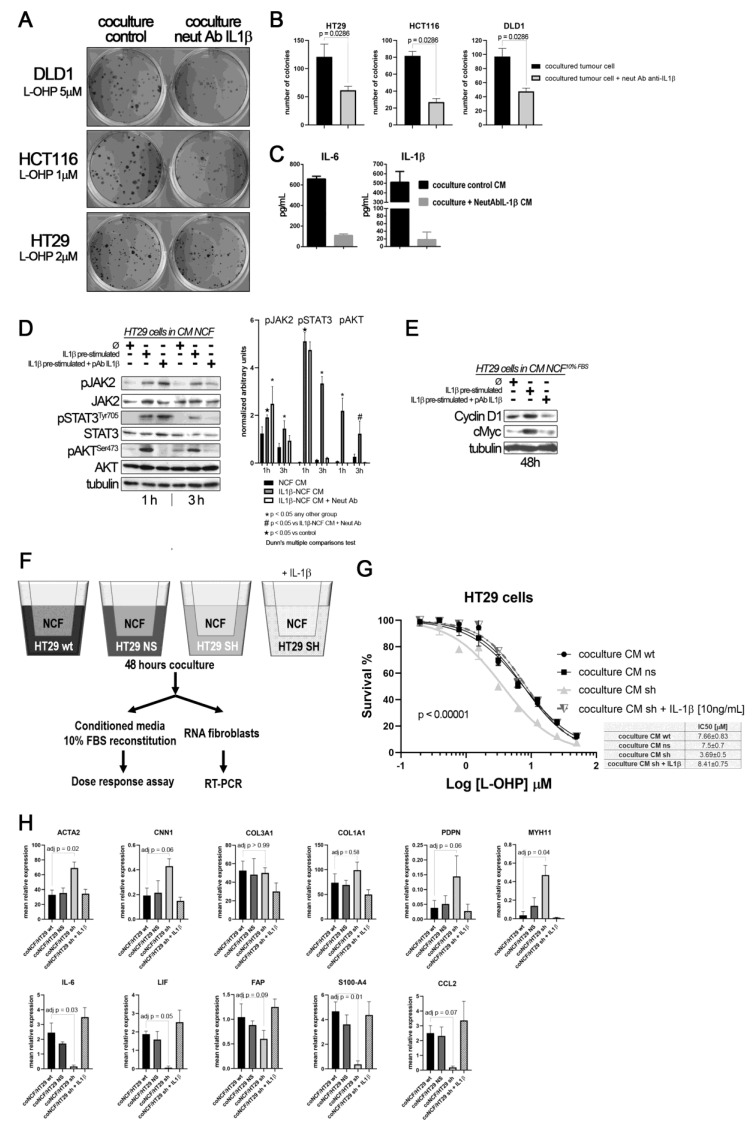
(**A**) colony forming assay of CCCL in transwell coculture with NCF. (**B**) quantification of colonies: Blocking the IL1β-mediated crosstalk between cocultures of NCF (upper 24 mm transwell chamber) and colorectal cancer cell lines (lower transwell chamber) with a neutralizing IL1β antibody sensitizes cancer cells to L-OHP. (**C**) Such IL1β blocking altered the composition of conditioned media (affecting IL1β targets), as illustrated in the bar graphs, where the neutralizing antibody affected the IL1β itself and IL6, as a surrogate marker of the IL1β response (grey bars), both soluble factors determined by ELISA in a mixture of coculture supernatants before 10% FBS reconstitution (proportional volume of the different CM from NCF with either DLD1, HT29, or HCT116 cells). (**D**) Western blot of HT29 cells cultured with control CM (Ø) or IL1β-stimulated NCF-conditioned medium (10 ng/mL of IL1β) or same condition with the addition of a polyclonal neutralizing antibody against IL1β (2 µg/mL). FBS-free DMEM/F12 was used to generate conditioned medium after 48 h NCFs culture with or without the presence of the neutralizing antibody. Such conditioned media were then used to stimulate JAK2, STAT3, or AKT in HT29 cells for 1 h or 3 h. Quantification of phosphoproteins for three independent experiments was performed normalizing first for total JAK2, STAT3, or AKT and then normalizing for Tubulin (data expressed as arbitrary units). Statistical significance was assessed using non-parametric Kruskal–Wallis + Dunn’s multiple comparison test. (**E**) In 48 h experiments, the same conditioned media were reconstituted at 10% FBS. We evaluated JAK/STAT target proteins, Cyclin D1, and cMyc. (**F**) Overview of the experiment: to confirm paracrine signaling mediated by tumor cell-derived IL1β, we cocultured NCFs and HT29 cells with a defective secretion of IL1β, (silenced by means of shRNA) or transfected with a mock vector or wild-type as controls (75 mm transwell inserts, 3µm pore-size). As a positive control, we added IL1β to cocultures with HT29-shIL1β and NCFs. Culture conditions were: 2 × 10^6^ cells tumor cells in the lower chamber and fibroblasts in the upper chamber (10^6^ cells) in FBS-free DMEMF12. After 48 h, we harvested the conditioned medium and reconstituted the 10% FBS. As illustrated in (**G**), the conditioned medium obtained from cocultured NCFs and IL1β-deficient tumor cells (Ht29shIL1β) yielded lower IC_50_ values in dose–response assays compared with the other experimental conditions tested (*p* < 0.0001; survival is reported as %). The conditioned media obtained from cocultures of NCF and HT29-shIL1β cells with the exogenous addition of IL1β restored the IC_50_ values of cocultured with HT29 wild-type cells. Each dose–response curve corresponds to the mean of three independent experiments of six technical replicates each. Differences between dose–response curves were compared with extra sum-of-squares F test (Log IC_50_). (**H**) Real-Time PCR of the aforementioned cocultured NCF’s reported that the inhibition of the IL1β-mediated crosstalk between HT29shIL1β cells and fibroblasts induced a myofibroblastic phenotype in NCFs, with increased expression of ACTA2, CNN1, PDPN, and MYH11, while inflammatory markers were diminished, evidenced by decrease in IL6, LIF, and CCL2 (Kruskal–Wallis test; adjusted P values after Dunn’s multiple comparison test).

**Figure 5 ijms-22-04960-f005:**
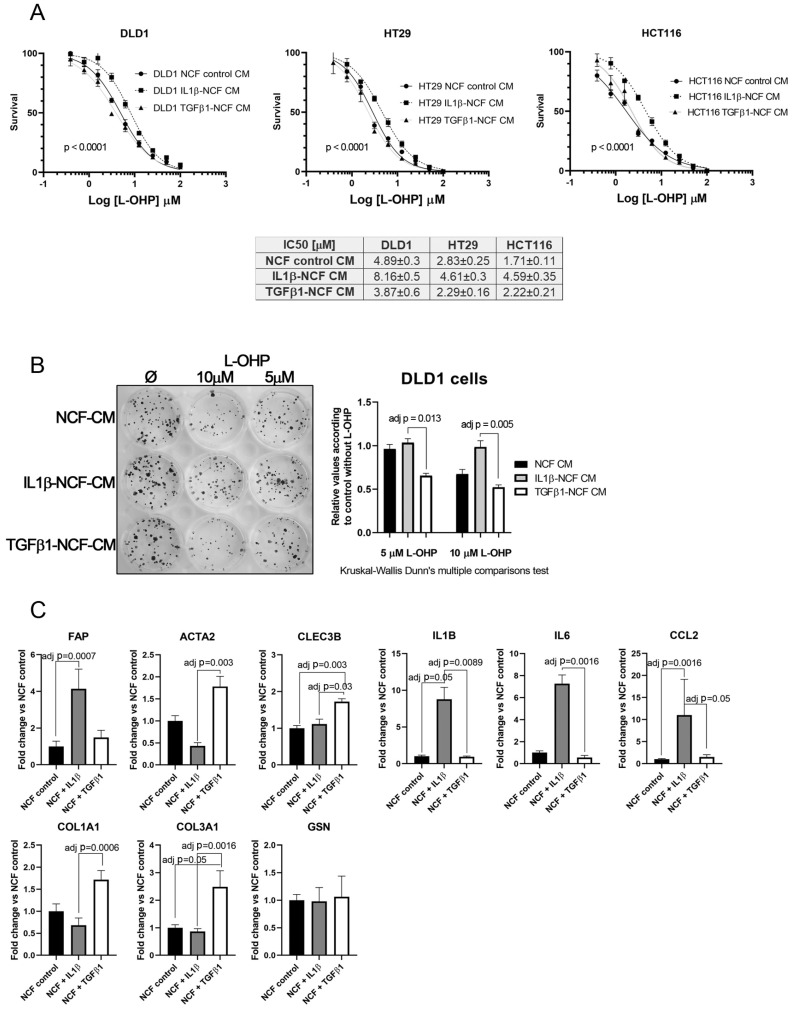
(**A**) Dose–response curves of L-OHP treated DLD1, HT29, and HCT116 cells cultured in NCF control conditioned media, IL1β-stimulated NCF conditioned media or TGFβ1-stimulated NCF conditioned media. For all cell lines tested, IL1β-treated NCFs conditioned media induced an increase in the IC_50_ values against L-OHP, while values for TGFβ1-treated NCFs media did not differ from NCF control conditioned media. Each dose–response curve corresponds to the mean of three independent experiments of four technical replicates each. Differences between dose–response curves were compared with extra sum-of-squares F test (Log IC_50_). (**B**) Colony forming assay of DLD1 cells (seeding density, 400 cells in twelve-well plates). We cultured cell lines with the aforementioned conditioned media in the presence of two different L-OHP concentrations. The quantification of the colonies reported that IL1β-treated NCFs conditioned media produced more colonies than TGFβ1-treated NCF’s conditioned media (Kruskal–Wallis test plus Dunn’s multiple comparison test, adjusted P values). (**C**) The expression values of different iCAF and myCAF markers were assessed by means of quantitative PCR in treated NCFs, showing that IL1β treated fibroblast acquired traits of iCAF, with the exception of CLEC3B and GSN, genes attributed to iCAFs in different publications [20]. Results expressed as fold changes in relation to normalized control.

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
