# Peer review of "The Blockade of Tumoral IL1β-Mediated Signaling in Normal Colonic Fibroblasts Sensitizes Tumor Cells to Chemotherapy and Prevents Inflammatory CAF Activation"

_ijms, 2021, doi:10.3390/ijms22094960_

Round 1
Reviewer 1 Report
The manuscript by Guillén Díaz-Maroto and collaborators describe the involvement of IL1B in the cross talk between colorectal cancer cell and fibroblasts. In this way IL1B induces the loss of a myofibroblastic phenotype in normal colonic fibroblast to stimulate iCAF traits. Moreover, different in vitro data support the induction of colorectal cancer cell chemoresistance mediated by IL1B. The manuscript present several in vitro experiment and provides interesting data in the knowledge of crosstalk communication between CCR cells and fibroblasts. Moreover, the manuscript is well written and easily read, particularly the discussion section. However, some issue should be improved before publication in IJMS.
Major points:
- Line 156-159: “Considering the migratory capabilities of cells, fibroblast migration was induced after IL1β stimulation, both, wound-healing assay (Figure 2F; P = 0.0035 Kruskal-Wallis test, with Dunn’s post hoc multiple comparison test) and directional migration (Figure 2H; adjusted P = 0.0034 Kruskal-Wallis test, with Dunn’s post hoc multiple comparison test), while DLD1 cells did not increase migration as expected (data not shown)”. If HT29 or HCT116 cells respond to IL1B, why did not test whether IL1B is affecting their migration?? I think that although any kind of results regarding this issue would not affect the chemoresistance data, additional data regarding this point could be interesting for the manuscript.
- Fig 4A and 4B: I think that the controls used for these experiments are not correctly set up. First of all, what are the effects of L-OHP in the fibroblasts? Any experiment should address this issue since if IL1B is mediating fibroblast proliferation (as authors previously confirmed), the effect of IL1B inhibitor on CRC cell chemoresistance may only be due to a reduction in the number of fibroblasts along with the action of L-OHP. To avoid this issue the experiments could be performed with conditioned medium of fibroblasts added to CCR cells incubated or not with L-OHP. But probably the most important point is referred to the assessment of the effect of fibroblasts/IL1B loop on CCR chemoresistance. The authors also need to show the effects of IL1B on CCR cells chemoresistance without any kind of coculture. I mean, CCR alone +/- IL1B and CCR+fibroblasts +/- IL1B and all of these conditions treated with L-OHP. Maybe this issue could be supported by the data showed in figure 2D and 2E in which the incubation with IL1B does not seems to affect L-OHP chemoresistance, but these data might be deeply analysed or at least discussed in the discussion section.
Minor points:
- Abstract: The authors stated: “We silenced IL1β in tumor cells to demonstrate that such cells do not exert an influence on NCFs inflammatory phenotype. IL1β is overexpressed in cocultured tumor cells”. Although this information is accurate, when reading the abstract it is a bit confusing. After reading the whole paper it is understandable, but the abstract should be informative, concise and accurate by itself. Please try to improve this issue.
- Fig 1A and 1B: Experiments with NCFs and CAFs are included, but then the whole study is developed only with experiments with NCFs and how they repolarize to an inflammatory or myofibroblastic phenotype. From this point of view, I consider that the data with CAFs do not contribute much to the work and should be eliminated.
- In a similar way, I do not understand why in some experiments (Fig 1H) are included normal hepatic fibroblasts or even 5-Fu experiments (Fig 3B). These data should be eliminated or commented in the text explaining why they are incorporated to the study and what they are adding to the work. In addition, why in experiment 3C author used for the experiment foreskin fibroblasts but they are not used in any other moment. And why are these fibroblasts used in this experiment?
- 2C: what is B10, B11… are they NCFs? Why this nomenclature just at this picture?
- Line 154: I guess it should be Figure 2E instead 2D.
- Figure 2I. I guess it should be cited in line 166.
- Line 280-283: “Similar results were observed in a colony forming assay, where IL1β -treated NCF conditioned medium significantly induced the formation more of colonies than TGFβ1-treated NCF…)” I think it is more correctly to state that “significantly avoid the reduction of colony” instead of “induced the formation more…”
- 4F: it seems that siRNA mock is also affecting some characteristics of CAF repolarization (fig 4H), thus I think that effects of IL1B with siRNA mock should be showed if possible.
- Fig 5A and 5B: It seems that TGFB treatment makes L-OHP more toxic to CCR cells, what could be the explanation for this? I think it would be good to comment these results in the discussion section.
Reviewer 2 Report
In this manuscript “The blockade of tumoral IL1β-mediated signaling in normal colonic fibroblasts sensitizes tumor cells to chemotherapy and prevents inflammatory CAF activation” Natalia Guillén Díaz-Maroto and collaborators describe the IL1β-mediated interplay between cancer cells and normal colonic myofibroblasts (NCFs). They found that normal resident fibroblasts become carcinoma-associated fibroblasts, conferring protumorogenic properties on these supposedly normal cells. They describe the IL1β-mediated interplay between cancer cells and normal colonic myofibroblasts (NCFs), which bestows differential sensitivity to cytotoxic drugs on tumor cells. NCFs and their conditioned media (CM) were used to characterize tumorogenic properties in cancer cells. Silencing of IL1β in tumor cells was used to demonstrate that cancer cells do not exert an influence on NCFs inflammatory phenotype. IL1β enables paracrine signaling in myofibroblasts, converting them into inflammatory CAFs (iCAF). In addition, compared with NCF-CM, ILβ-stimulated-NCF-CM induces migration and differential sensitivity to oxaliplatin in colorectal tumor cells. IL1β induces the loss of a myofibroblastic phenotype in NCFs and acquisition of iCAF traits. The authors concluded that IL1β confer protumorogenic features on them, particularly de novo resistance to cytotoxic drugs.
In general, the manuscript is interesting and the authors tried to address a rilevant issue.
However, the authors should address the following comments/questions:
- In general, all the data are based just on in vitro Did the authors consider to use patient-derived fibroblast in vivo experiments?
- The effects of CAF on drug resistance are based on experiments that used only L-OHP. It is recommended to test author’s hypothesis using more than one drug.
- The manuscript contains several grammatical/ syntax and requires an extensive English stile revision. For example: line 177, line 280.
- Fig 1: IL1beta protein expression by WB is required.
- Fig 2: the authors tried to analyse the impact of IL1b on CRC and Fibroblast proliferation. Then they switch migration analysis. Why? Please explain. This part of the manuscript should be revised.
- Fig 2C: Why a P38 inhibitor VX-702 (400nM) has been used? Please, explain.
- Fig 2G: To prove that DLD1 migration is IL1b-mediated, cells should be pre-treated with blocking anti-IL1b.
- Line 156-160 sentence is not clear. Please, re-phrase.
- Line 160: “while 160 DLD1 cells did not increase migration as expected (data not shown).” The effect of IL1b on DLD1 should be shown.
- It has been reported that inflammatory niches consisting of tumor-associated macrophages and fibroblasts contribute to tumor drug resistance through a cytokine-signaling network that involves macrophage-derived IL-1β and fibroblast-derived CXCR2 ligands. In particular, fibroblasts require IL-1β to produce CXCR2 ligands, and loss of host IL-1R signaling in vivo reduces melanoma growth (Young H et al J Exp Med(2017) 214 (6): 1691–1710.). This should be included in Discussion.
- Fig 3C. HCT166 looks to be resistant. Are the authors sure that is not due to an increased in proliferation (see Fig 2B). Did the authors test it in the presence of anti-IL1b?
Reviewer 3 Report
In this manuscript, the authors investigate the role of IL1 β signaling in fibroblasts on mediating differential sensitivity of cytotoxic drugs in cancer cells. IL1β signaling transforms the normal fibroblasts to inflammatory CAFs that modulates the migration and differential sensitivity to oxaliplatin in colon cancer cells. While the effect of Il1β transformed fibroblasts on tumor cells is evident in the experimental readouts, the study design is not appropriate for the CM transplant experiments. The CM is prepared in presence of IL1 β cytokine that does not rule out if the observed effects are due to fibroblast secreted factors or due to direct effect of the cytokine. The study does provide insights into the role of iCAFs in regulating sensitivity of colon cancer cells to oxaliplatin, however authors should address the below mentioned comments before the manuscript is considered for publication:
Major comments on the manuscript:
- The authors have used CM generated from NCFs to study the effect on colon cancer cells. The conditioned media is generated in the presence of IL1β. The authors should pre-treat the cells with IL1 β and then generate CM to show the effects on the cancer cells. Also, the method section describes a time of 24hrs for CM generation while in results 48hr is described. The discrepancy should be resolved.
- The study lacks rigor in the choice of cell lines. Strangely, some of the experiments are done with HT29 cells and some are performed with DLD1 cells. The authors must provide data for at least these two cell lines.
- Two knockdown constructs are described in the method section while only one is used for the study. It is advisable to use at least two knock-down constructs. The knock-down construct details should be provided at appropriate places.
- Are the multiple NCF cell lines used in the study characterized? The details should be included in the methods section.
- The IC50 values should be reported for the L-OHP dose-response curve in the Fig 3A, 3C, 4G and 5A.
- How are the values normalized in the Fig 1H. The exact method of normalization should be described in the figure legend. Can the authors explain why no difference is observed in the endogenous levels of Il1β receptors between fibroblasts and cancer cells?
Minor comments:
- The phospho-signaling blots should be quantified and reported for Fig 4D.
- The raw data for all the detected proteins should be provided in the supplementary table.
Round 2
Reviewer 1 Report
The manuscript was improved and I think that now it should be accepted for publication in IJMS.
Author Response
Grateful to have been able to discuss and enrich the manuscript
Reviewer 2 Report
In general, the authors addressed all the issues. The manuscript is improved.
Author Response

(The authors gave the same response as above.)

Reviewer 3 Report
The authors have clarified in the response that the CM is prepared from IL-1β treated cells after 48 hrs. I presume the cells were washed after 48hrs of treatment and CM was prepared from these treated cells. If yes, they must mention this in the methods section as it is still not clear from the text. If the CM generated is free of IL-1β, it provides significance to the findings. With this background, the manuscript can be considered for publication if the authors can provide justification for the following comments:
- The effect of IL-1β treatment on cell lines should be investigated in multiple cell lines. While it is understood there is heterogeneity in cell line physiological properties. The use of single cell line dilutes the significance of the findings. It is strongly recommended to use at least two cell lines for validating the functionality or at least a rationale should be provided for such observed differences.
- As suggested in the previous review, only one shRNA was used to target IL-1β in the cells. If the shRNA is a limitation, the authors must validate their results using sgRNA for IL-1β.
Round 3
Reviewer 3 Report
The comments and suggestions provided in the previous version of the manuscript were intended to improve the quality of the manuscript and authenticity of the conclusions. However, considering the justifications provided by the authors, the manuscript may be accepted in the present format.